# Ready Player Viz: Player-Created Strategic Visualizations for Video Games

Jane L. Adams*    Michael Davinroy†

Khoury College of Computer Sciences
Northeastern University

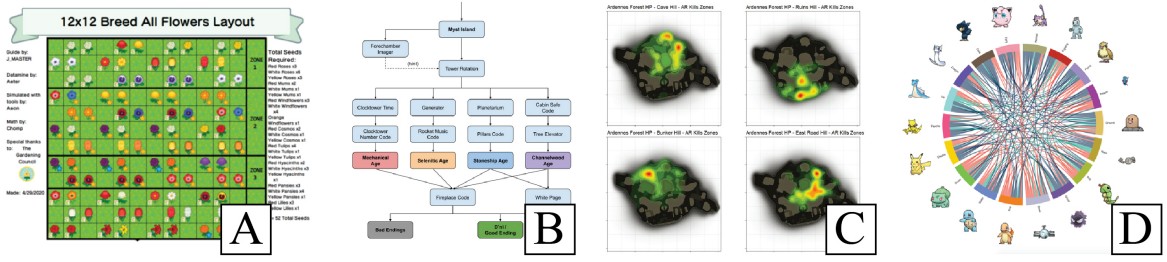

Figure 1: Some examples of visualizations from the survey: A) A plant hybrid breeding matrix diagram from *Animal Crossing: New Horizons*. B) A node-link flow chart diagram posted by a player of puzzle game *Myst*, shared to demonstrate a problem with the existing user flow and a proposed solution to improve gameplay. C) Small multiple 3D heatmaps from a player of *Call of Duty*, showing "AR kills... on all four Ardennes Forest Hardpoint hills throughout every WWII Major LAN". D) A chord diagram of *Pokémon* interactions built by a player of the game using D3.js.

## ABSTRACT

We present a survey of the design space of player-created strategic visualizations for video games, to better understand how players make sense of complex game mechanics and incorporate feedback from fellow players. We present several examples of these visualizations, and contrast them to developer-created visualizations, both in information presentation and purpose. We find that there is a rich community-building aspect to visualization development within game 'fandoms', facilitated by cross-platform exchange and iterative development, including: social media, informational 'wikis', and in-game modifications ('modding'). Finally, we consider player-created visualizations in the context of a broader imagining about the future of visualization development for non-game but analogous strategic applications. We maintain a collection of tagged and categorized examples of player-created video game visualizations at https://dev.universalities.com/playerviz.html.

## 1 INTRODUCTION

There exists a wealth of information about user interface design within video games [8, 12], player behavior within games [9], and analysis of game mechanics from an academic perspective [7, 10]. However, an area that is presently under-explored is the visualization underworld of *player-created* data visualizations for understanding and optimizing gameplay. We have performed a preliminary survey of this design space by collecting 89 different visualizations from various gaming communities and categorizing them by visual encoding type. In this process, we have unearthed some interesting characterizations of the player-created visualization design space, and posit that they may provide useful information for the broader visualization research community.

---
*e-mail: adams.jan@northeastern.edu
†e-mail: davinroy.m@northeastern.edu

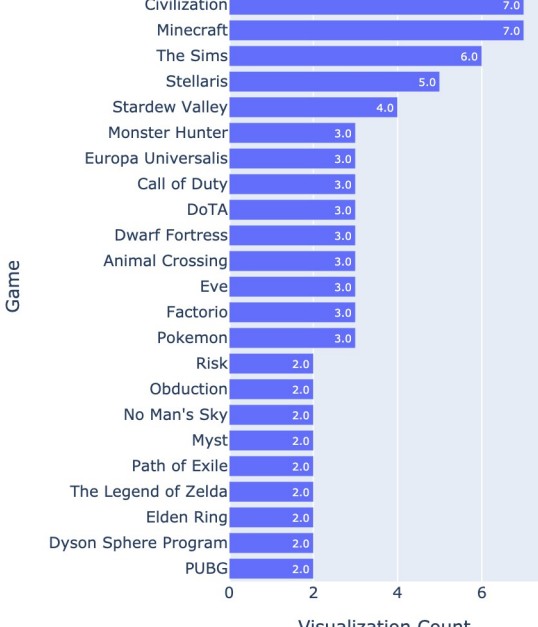

Figure 2: Top games represented in the survey data (where count exceeded 1), by number of visualizations present. In total, there were 38 games in the survey and 89 total visualizations. Of the initial game communities contacted, at least one community member responded in approximately 2 out of every 3 requests, but additional games were included thanks to members of contacted communities suggesting other visualizations, creators, and games.

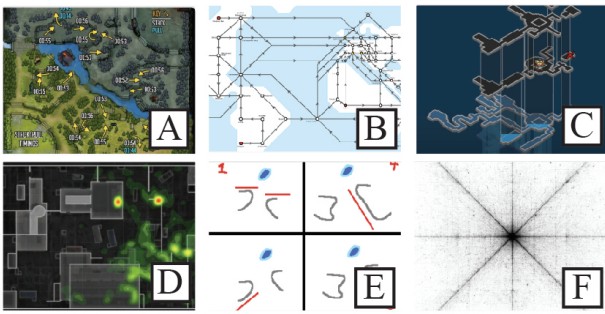

Figure 3: Examples of map visualizations from the survey. A) A "Stack and Pull" guide for *DoTA* showing when to multiply neutral 'creep' spawn locations ("stacking") or bait normal creep waves to neutral creep camps "pulling" to lessen the wins of the enemy. This annotated visualization is overlaid over the ordinary in-game minimap. B) Subway-style map of the global trade routes in *Europa Universalis IV*, a strategy game based around trading. Edges are directed in this node-link diagram, and nodes are colored as trade nodes, source nodes, and end nodes. C) Isometric rendering of an achitectural blueprint for a multi-level dungeon in *The Legend of Zelda*. This visualization accompanies a YouTube video from the "Boss Keys" series, which teaches players how to defeat particularly challenging opponents. D) Heatmap from *Call of Duty* overlaid on an architectural blueprint of a particularly deadly location in-game. According to its creator, this location, "Hackney P2", "led all hills in average score difficulty for rotation winners" and the heat visualization shows "deaths during first 30 seconds of break attempts from past couple months of scrims" (scrimmages). E) Boss location map from "Path of Exile", this simple visualization is actually a set of 4 small-multiple maps, created in Microsoft Paint for the purposes of conveying the statistically probable locations of enemy spawn locations. F) Dasymetric dot density map of player locations from *Minecraft* server 2b2t, a server "world that is 11 years and 2 months old (as of July 2022), with a size of 14,198 GBs and over 733,661 players having visited at least once". We can see from this visualization that players congregate in the center of the world, and then move outwards, often at 45- or 90-degree angles from the center location.

## 2 RELATED WORK

It's no secret that video games, while they foremost exist for entertainment purposes, have significant utility for broader understanding of the world. For example, the 2005 "Corrupted Blood" pandemic bug in *World of Warcraft* [3, 6] made waves in the epidemiological research community for understanding how humans respond to pandemics, e.g. medics ("healers") swarming into infected areas to attempt to save their friends, only to re-infect others when the medics returned to hubs to resupply. The MineRL Competition was started in 2019 as a testbed for deep reinforcement learning in a challenging, sparse rewards environment, using human data in the open game world of *Minecraft* [4]. The results of this annual competition inform research understanding of model design and reward function specification. Sometimes, interactive media even blurs the line between game and lesson, as with flight simulators, for which there exist both entertainment-focused users as well as aircraft pilots, for whom flight simulators consistently contribute to improvements in flight training [5]. Therefore, it is useful to consider how user-created improvements to gameplay might inform subsequent improvements to real-world problem spaces.

While video games arguably exist in a larger space of categorization, arts and technology researcher Veronica Zammitto organizes video games into three key genres: First-Person Shooter (FPS), Real-Time Strategy (RTS), and Massively Multiplayer Online (MMO). She notes that FPS games use the point of view (POV) of the user's

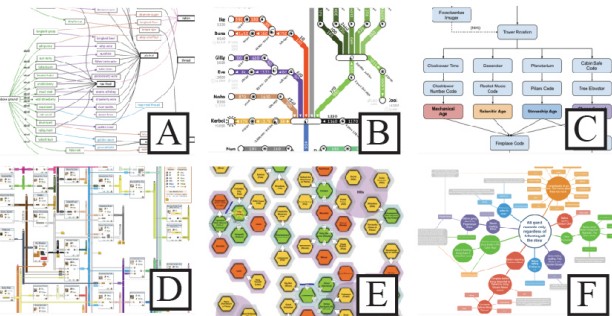

Figure 4: Examples of node-link diagrams from the survey. A) A diagram of crop relationships from the game *Dwarf-Fortress*. B) A subway-map style flow visualization of different Δ velocities needed to traverse various in-game celestial bodies in the game *Kerbal Space Program*. C) A tree representing the dependencies of puzzle solutions in the game *Myst*. D) A flow of resource production and usage in the game *Workers and Resources*. E) A hex grid showing mandatory adjacencies for wonders and optimal district adjacencies in *Civilization VI*. F) A network graph showing quest rewards that might be missed if other certain game conditions have already been fulfilled in the game *Elden Ring*.

avatar, and thus frequently employ user interfaces such as heads-up displays (HUDs). RTS games, on the other hand, often show maps or schematics, and involve dashboards for much of the gameplay, featuring minimaps, buttons with selective coloration for pre-attentive processing, and pop-out effects. MMO games commonly feature chat windows for coordination with teammates, and a POV that is either first-person or first-person but embodied above the character so the avatar can be seen navigating the environment. Zammitto notes that MMO games are often peppered with icons, and spend a lot of visual space on character development and inventory management. She concludes her literature review with this call for further research: "Information visualization in digital games is still not very developed. The main body of work on this topic comes from the game developer industry, but the lack of an established terminology for the field makes it difficult to work towards its foundations. Looking into the established information visualization field would help in this regard" [12]. It is in this spirit that we chose to investigate the other side of this question: what visualizations do *players* create to embellish, augment, or supplant the in-game visual design systems?

## 3 METHODS

As players ourselves, we had some examples which inspired this project initially, which we used in a collaged image to solicit additional responses from game communities. We chose game communities based on a combination of game popularity, number of active members on Reddit, and prevalence of player-created visualizations in prior posts. In our solicitations, we stated, "Hey [X] community! I'm a student researcher looking at player-created data visualizations for video games and board games. Do you have any favorite \*player-created\* data visualizations from your community to share? Here are some cool examples I've found from other game communities" followed by the collage image. We learned that posts needed to be tailored to each subreddit in order to evade moderation bans, e.g. by tagging "No Spoilers", posting an "R5 comment" describing the image (*Rule 5: If you post a screenshot of the game, please point out what you want people to look at in the image or explain in the comments.*), or waiting until certain days of the week when user-generated content or images were permitted. Some communities were unresponsive (approximately 1 in 3 that were contacted), but many were excited to share examples. Interestingly, several users referred us out to other communities that we had not considered: *EVE*

*Online* and *Factorio* were two in particular that were recommended by users on several occasions.

Following the user recommendation period, we collected all of the responses (some of which required additional digging to find, since links were omitted), and added them to a large survey document. We also sourced some additional visualizations by querying within Fandom Wikis and subreddits for terms like "[game] cheat sheet","[game] data visualization", and "[game] chart". We collected a title for the visualization, the game or game community from which it came, the image(s), the source link, and the name of the referrer (if applicable). 'Title' in our survey data may be the user- or referrer-assigned title, a shortened version of this title, or, for visualizations with non-descript or non-existent titles, we have attempted to assign an informative title from the content and context of the visualization. Additionally, we hand-selected at least one chart type categorization for each visualization. We relied on chart types from the Data Visualization Catalog (`https://datavizcatalogue.com`) where there was ambiguity, and also applied multiple categorizations to complex or unusual visualizations. We do not distinguish here between visualizations that use elements of the base game (e.g. maps and skill trees), because of certain ambiguities, such as visualizations using iconography from the game but novel orientations of data. We did omit repeated visualizations of the same type: for example, Figure 3 (A) from DoTA was from a website with several pages of strategy maps; we chose a single visualization as an example to include so as not to over-inflate the representation of a single game or visualization type.

## 4 FINDINGS

We organize our findings into our observations about the iterative visualization design process as evidenced by community forum contexts in which visualizations were found, and typology of visualization, for which many visualizations were assigned multiple categorizations.

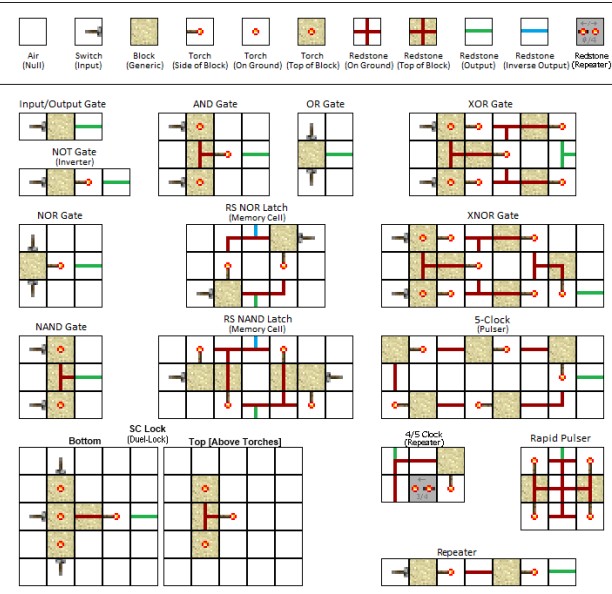

Figure 5: Small multiples visualization of building logic gates in *Minecraft* using Redstone.

### 4.1 Community-Driven Design

Expert-driven design is often a hierarchical, top-down design process in which a 'patron-as-client' leads the development of a product based on the needs of stakeholders. This is likely the typical development workflow within game development teams, wherein in-game visualizations are designed to advance the specific intended ludic goals of the game mechanic. In contrast, Community-Driven Design finds roots in the needs of collective group, who push forth solutions to problems the designers (players) themselves experience. While this type of design has roots in the physical world and the design of communities, digital spaces and the internet have pushed this boundary to a larger scale, from open source software to community forums [11]. In particular, we examine community forums, such as the Steam Community Forum, Reddit, and Fandom, that exist as rich resources for players to solicit (or, in some cases, receive unwanted) feedback on their visualizations of game strategy. For example, consider Figure 5, and the following comments from users:

- "Aesthetically I've always hated that diagram. I've taught circuitry and used diagrams like this a lot and found that students new to the subject learn better off of 3D renderings of the circuit." - u/Adolpheappia

- "A tad out of date, what with the new piston systems, but agreed that is a very handy diagram" - u/greentrafficcone

- "I actually find the diagram more confusing than someone telling me how to make it." - u/iPeer

In addition to these comments describing the user experience of making sense of the charts, there is detailed conversation about how to improve the visualization, information that is missing, and real-world analogies to electrical engineering circuit design. In our survey, we have included links to the source material for each of the visualizations surveyed not only as a typical citation, but also as a further resource because many of the visualizations posted boast long comment threads of community input and refinement. However, we have opted not to archive Reddit threads directly due to consideration of the 'right to be forgotten' [1]. Deleted threads can still be accessed by using the metadata in the URLs with the PushShift API, but users whose data is archived there retain the ability to opt to be removed from the PushShift database [2].

In many cases, users create visualizations knowing that there will be community feedback, and apply version control systems to their design process. As shown in Figure 6, edits are made iteratively to a hex map showing adjacency bonuses of districts in the turn-based strategy game *Civilization VI*. From a visual encoding perspective, we can consider this to be both a map and a node-link diagram, wherein adjacency bonuses are edge weights, and nodes are districts. Coloring is applied to nodes based on their district type.

### 4.2 Typology

After collecting all survey data, we ran some data analysis on the games and visualization types represented in the data. Games that were included more than once are shown in Figure 2, and visualization types that were found more than once are shown in Figure 7.

The most common visualizations found in the survey were Node-Link Diagrams (30 of 89). This is most likely due to three main types of node-link visualizations represented in the data:

- **1. Recipes** These types of visualizations demonstrate how certain game components combine into other components, e.g. how herbs might combine into a potion, or how raw materials can be smelted into weapons.

- **2. Timelines** Many temporal maps employ node-link diagrams to outline branching narratives, such as "tech trees" in RTS games, or "quest lines" in MMO games.

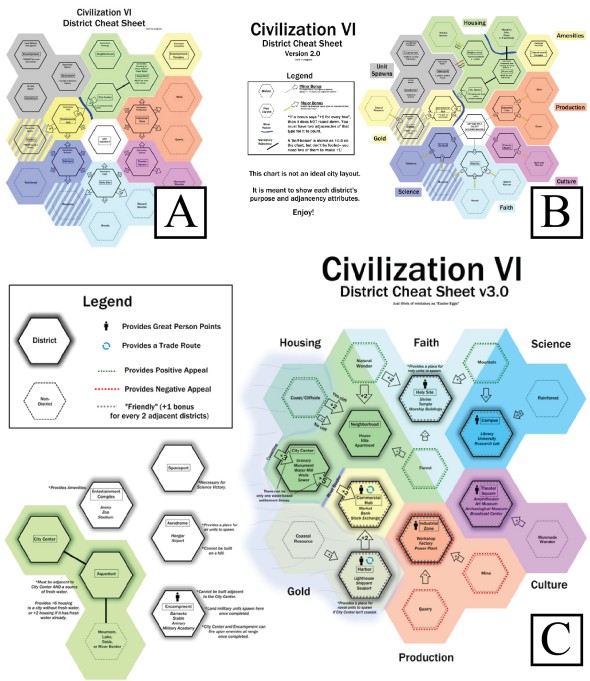

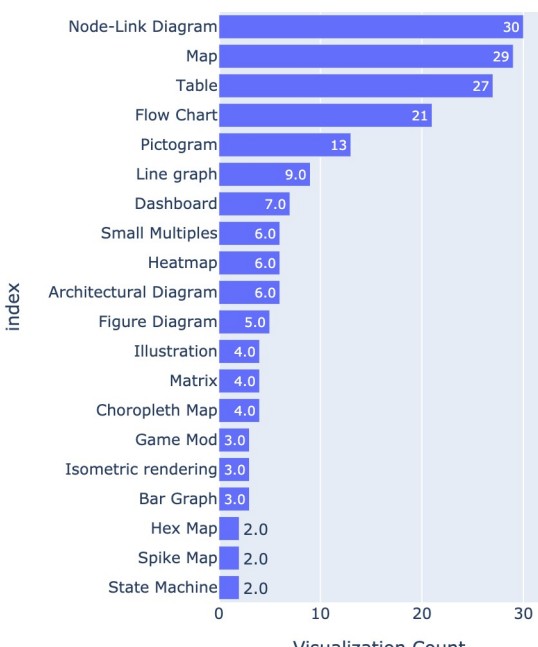

Figure 6: Three iterations of hex maps showing the adjacency bonuses of districts in *Civilization VI*, created by reddit user u/iotafox in 2016. Version A) had some issues, e.g. "aerodrome (spelled wrong on chart) can't be on a hill" - u/redaelk. Version B) was posted soon after, with further edits, e.g. "Correction: In +1 for every 2 it says 'Does not round down' when it should say 'Does not round up'" - u/The_KazaakplethKilik. Version C), shared with the comment "Just think of mistakes as 'Easter Eggs'", comments are mostly just "this is really well made" - u/coltblood, and "After 3 editions this is almost perfect. Kudos, greek for glory, to you my friend. I will be gladly using this in my first game in about 10 mins." - u/ProudNitro.

- **3. Maps** Often, complex maps are represented more simply as node-link diagrams in games where physical distance is less important than connectivity. This is especially common in games with trade routes, where players are traversing a large map, such as a star system, and are more concerned with connectivity between points rather than terrain.

The second most common visualization found in the survey was maps. Across many of the games surveyed, maps were commonly generated by players both for strategic purposes as well as narrative goals. Maps ranged from small-scale, such as blueprints of dungeon layouts, to world- and even galaxy-scale. Some player-created maps were simply annotation layers applied to in-game maps, whereas others were completely new creations, some even hand-drawn.

The third most common visualization found in the survey was tables. We hypothesize that tables are in part most common because they are the easiest for novice visualizers to create. Many of the tables we found in the survey used simple tools like Google Sheets or Microsoft Excel to coalesce complex data about in-game interactions.

Some of the more surprising visualization types that we found in our survey included a dasymetric dot density map, a 3D "spike map" of density, two examples of state machines (a special kind of node-link diagram), and even a chord diagram showing relationships between *Pokémon* types, which was created using D3.js.

Figure 7: Top visualization types (where count exceeded 1), represented in the survey data by number of visualizations present. In total, there were 30 different visualization types in the survey among the 89 total visualizations. Note that some visualizations had multiple chart types listed.

### 4.3 Future Work

This work could be continued by snowball sampling further into game communities or publicizing the project on other platforms beyond Reddit, such as Twitch or YouTube. Additionally, for the bulk of these visualizations, it is unknown *how* they were created (e.g. which software was used, or whether there were preliminary pen-and-paper sketches). A follow-up ethnographic study would answer some of these questions and provide a window into the amateur visualization artist's toolkit. This might also bring attention to gaps in the available tools for first-time visualization designers that could be solved by 'what you see is what you get (WYSIWYG)', code-free tools, perhaps specific to certain games (e.g. providing base hexagonal maps for *Civilization* cartographers, or APIs for reading game data directly into a plotting library).

### 4.4 Conclusion

We are excited to present our preliminary survey, and hope that others will continue to make recommendations to add to the library so that this data set can inform future research about player-created visualization types. Additional studies could be performed related to the relationship between players and game developers; the tools that players use to create these visualizations; and the differences in chart type prevalence across different game type categories.

### 5 ACKNOWLEDGEMENTS

Thank you to Star St. Germain, and the Reddit users listed in the corresponding items in the survey data, for recommendations of examples 'in the wild' to look at. Thanks also to Sara Di Bartolomeo for encouragement and feedback, and to the alt.VIS reviewers for taking the time to read our work.

## 6 DECLARED CONFLICTS

Jane Adams is an organizer of alt.VIS 2021 and 2022.

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
