# OpenReview forum: "Ready Player Viz: Player-Created Strategic Visualizations for Video Games"
_IEEE.org/2022/Workshop/altVIS — Accept_

### Official Review · Reviewer_hJfr · 2022-08-05

**Review:**


Is it interesting? Yes.

Is it weird? I'm not sure - it's currently a bit short, but I don't think there's anything about the topic or presentation format that would prevent an extended version from being accepted as a conventional conference paper.

However, the topic is a bit unusual, and I don't think it would be seriously out of place at Alt.VIS, even if could have been presented somewhere else instead.

This is an interesting survey that is clearly written; I don't know of any previous surveys in the same area.

## Minor comments:

- the workshop website suggests that there needs to be a COI statement, since Jane is an organizer

- the URL at he the end of the abstract should be an actual clickable hyperlink (see the LaTeX href package)

- consider italicising the names of games throughout

- I have no idea what an "R5 comment" is, and I suspect that I'm not alone in my ignorance

- how many game communities "were unresponsive"? It would be interesting to state this number in the caption of Figure 2

- it would be helpful for the reader if the labels ("A)", "B)", et.c) in the Figure captions were links to the corresponding entry on the supplementary website

- the article states that the authors "collected a title for the visualization". This is a little vague about who write the title: the creator of the visualization, the person who forwarded an image to it, the author of the paper... As a concrete example, one entry in the online catalog has the title "Cat Raid DPS Flowchart", which is slightly different to the text in the image ("Simplified Cat Raid DPS Flowchart")

- it would be interesting to know what tools were used to produce the visualizations, but I recognise that this may be difficult to determine in many cases.

- the paper states that the URLs of discussions were recorded "as a further archival resource", but unless these are actually archived they may succumb to link-rot. I suggest using a tool like arhiveror [1] or archivenow [2] to ensure they are recorded in public archives.


## Suggested edits to text:

- "we as authors" -> "we"
- "active members on Reddit" -> "number of active members on Reddit"
- "recieve unwarranted" -> "receive unwanted"
- "for game strategy" -> "of game strategy" (?)
- "Fig. 5": the space before the 5 looks a little large - perhaps you used "." rather than ".\" so that LaTeX interpreted the period as the end of a sentence
- "in turn-based strategy game" -> "in the turn-based strategy game"
- "The most common visualizations found in the survey were Node-Link Diagrams" -> "The most common visualizations found in the survey were Node-Link Diagrams (30 of 89)"
- "visualizations that were found more than once" -> "visualization types that were found more than once"
- "shown in 7." -> "shown in Fig. 7."


[1]: https://github.com/rahiel/archiveror
[2]: https://github.com/oduwsdl/archivenow

**Conflicts:**

None

**Review Inclusion:**

Yes

**Sufficiently Alt:**

Yes

---

### Official Review · Reviewer_CBuk · 2022-08-10

**Review:**

I like the premise of the paper. I think it is worth exploring what people who don't necessarily have a visualizations background come up with and what techniques they use. It would be interesting to know the general knowledge of the creators about visualizations.

The authors could've provided a distinction between visualizations made from scratch and those that use existing game elements, such as maps or skill trees, as the base.

**Conflicts:**

None.

**Review Inclusion:**

No

**Sufficiently Alt:**

Yes

**Superlative:**

Most playful.

---

### Official Review · Reviewer_2SSP · 2022-08-24

**Review:**

The research elaborate on the community created strategic visualizations in the video game compared to developer created video games and further extend the discussions to a broader non-game strategic scenarios.
Pros:
1. the research questions on community created strategic visualizations is innovation
2. the authors also provide a supplementary collection of visualizations: https://dev.universalities.com/playerviz.html

Cons:
the authors should include more discussion on existing literature and visualizations of community driven design: https://link.springer.com/chapter/10.5822/978-1-61091-893-0_2

**Conflicts:**

NA

**Review Inclusion:**

Yes

**Sufficiently Alt:**

Yes

**Superlative:**

Most community driven

---

### Official Review · Reviewer_L158 · 2022-08-31

**Review:**

Meta Review:
In this paper, the authors survey visualisations created by players to elaborate strategies. Reviewers agree it is a very interesting paper and that it should be presented at AltVis. We decided to accept it.

**Conflicts:**

No Conflict

**Review Inclusion:**

No

**Sufficiently Alt:**

Yes

**Superlative:**

Most strategicly playful

---

### Decision · Program_Chairs · 2022-08-31

Accept